# Adolescent Idiopathic Scoliosis Surgery: Postoperative Functional Outcomes at 32 Years Mean Follow-Up

**DOI:** 10.3390/children11010052

**Published:** 2023-12-30

**Authors:** Giuseppe Barone, Fabrizio Giudici, Francesco Manzini, Pierluigi Pironti, Marco Viganò, Leone Minoia, Marino Archetti, Antonino Zagra, Laura Scaramuzzo

**Affiliations:** 1Spine Surgery Division 1, IRCCS Ospedale Galeazzi—Sant’Ambrogio, 20157 Milan, Italy; giuseppe.barone@grupposandonato.it (G.B.); fabrizio.giudici@grupposandonato.it (F.G.); leominoia@tin.it (L.M.); marino.archetti@grupposandonato.it (M.A.); laura.scaramuzzo@grupposandonato.it (A.Z.); 2Residency Program in Orthopedics and Traumatology, University of Milan, 20122 Milan, Italy; francesco.manzini@unimi.it (F.M.);; 3IRCCS Ospedale Galeazzi—Sant’Ambrogio, 20157 Milan, Italy; marco.vigano@grupposandonato.it

**Keywords:** scoliosis, spine surgery, fusion, functional outcome

## Abstract

Introduction: Recent clinical and radiographic studies conducted over short and medium terms have demonstrated positive results in patients undergoing surgery for adolescent idiopathic scoliosis (AIS). However, the absence of long-term data, crucial for comprehending the impact on future quality of life, especially in young patients actively involved in very intense physical activities, remains a gap. This study aims to evaluate long-term functional outcomes in patients who underwent surgery for Adolescent Idiopathic Scoliosis. Material and Methods: Patients meeting specific criteria (diagnosis of AIS, age at surgery between 12 and 18 years, and follow-up of at least 20 years) were identified from a large spine surgery center database. A questionnaire using “Google Form” assessed various outcomes, including Visual Analog Scale (VAS) back, VAS leg, Short Form 12 score (SF-12), Scoliosis Research Society 22 score (SRS-22), incidence of spine revision surgery, postoperative high demanding activities (work and sport), and possible pregnancies was sent to the enrolled patients. The authors analyzed the results regarding all patients included and, moreover, statistical analysis categorized patients into two groups based on the surgical fusion performed: Group 1 (non-instrumented technique according to Hibbs–Risser) and Group 2 (instrumented tecnique according to Cotrel–Dubousset). Results: A total of 63 patients (mean age 47.5 years) were included, with a mean follow-up of 31.9 years. Patients were, in mean, 47.5 years old. Group 1 comprised 42 patients, and Group 2 had 21 patients. Revision surgery was required in 19% of patients, predominantly for implant issues in Group 2 (11.9% vs. 33%, *p* < 0.05). Overall outcomes were favorable: VAS back = 3.5, VAS leg = 2.5, SRS-22 = 3.5, SF-12 Physical Component Summary = 41.1, SF-12 Mental Component Summary = 46.7, with no significant differences between the group 1 and group 2. At 5-years FU, the non-reoperation rate was higher in the non-instrumented group (97.6% vs. 71.4%, *p* < 0.001). By means of SRS-22, overall satisfaction was 3.7 ± 1.2 on a maximum scale of 5. More than half of women have successfully completed one pregnancy. Most patients (87.3%) maintained regular work activity. Among sport practioners, half returned to the similar preoperative level. Conclusions: This study reveals favorable long-term functional results in adolescent idiopathic scoliosis patients after surgical fusion. Mild to moderate back and leg pain were observed, but overall satisfaction, sport participation, and work activity were high. Surgical technique (non-instrumented vs. instrumented) did not significantly impact long-term results, though the instrumented fusion exhibited a higher revision rate.

## 1. Introduction

The current literature suggest that adolescent idiopathic scoliosis has an estimated prevalence of 0.47–5.2% [1], using a cutoff point of 10° Cobb or more. AIS develops at the age of 10–18 years and the incidence for females is 1.4–2.1 times higher than for males. Survival analysis assessed that 0.7–1% of diagnosed patients underwent surgical treatment within five years. Surgery was most frequently performed at 12–14 years of age [2].

Whether or not an AIS patient should undergo surgical intervention depends on several factors including the overall curve size and pattern, curve progression, and skeletal maturity. Surgery is considered in skeletally immature patients with structural curve Cobb angles over 40° [3]. The natural evolution of adolescent idiopathic scoliosis (AIS) is unclear, especially for Cobb angles between 30° and 60° in adolescence. Historically, there is no clear consensus on the exact cutoff for scoliosis surgery, several factors including the overall size and pattern, curve progression, and skeletal maturity. In the recent past, a cutoff was set at 50° with a progressive reduction to 40° [3]. Otherwise, surgery is also recommended in immature patients with a progressive structural curve in the last six months of observation, in which important residual growth is expected.

Correction and fusion surgery have been used for the treatment of scoliosis since the early 1900. Russell Hibbs performed the first scoliosis fusion by posterior open release and uninstrumented “in situ” fusion with subsequent prolonged cast-immobilization from 6 to 12 months. Driven by the desire to increase the amount of deformity correction rate and to reduce the nonunion rate, surgical techniques including the use of internal instrumentation were investigated and developed over the years. Harrington rods (1960s), Luque sublaminar wires (1970s), laminar hooks, and pedicle screws by Cotrel and Dubusset (1970s, 1980s) have been used to present.

Today, patients can be treated with different surgical approaches: anterior spinal fusion, posterior spinal fusion, or a combined approach. It is estimated that currently 75% of AIS surgery is performed with a posterior-only approach [4]. New implants, new surgical approaches/techniques, and modern technology result in better surgical outcomes.

Since most of patients affected by scoliosis undergo surgery at a very young age, it is important to know long-term results, especially with regard to clinical outcomes and future quality of life. Many young patients carry out activities with high-functional demand, such as sports or work, and for this reason both patients and their parents are very interested to obtaining excellent long-term functional results. Today, in the literature there are mostly short- and medium-term follow-up. The purpose of this study is to evaluate the long-term functional outcomes in young patients who underwent surgery for adolescent idiopathis scoliosis, paying special attention to pain, social and sport activities, overall satisfaction, and quality of life. Another aim is to assess if the first instrumented fusion techniques lead to better long-term results than the older non-instrumented fusion technique and to find out possible differences in revision rate.

## 2. Materials and Methods

This retrospective cohort study was conducted at IRCCS Ospedale Galeazzi—Sant’Ambrogio, Milan (Italy), Spine Surgery Division 1.

A database of 509 patients who underwent surgical treatment for scoliosis by our spinal surgery division from 1980 to 2001 was analyzed. The database included the following information: patients’ identity, diagnosis, age at surgery, other pathological conditions, date of surgery, detailed surgical procedure performed (extension of fusion, non-instrumented, or instrumented posterior technique).

We included patients diagnosed with AIS, age at surgery ≥ 12 and ≤18 years, undergoing posterior surgery, and with a follow-up greater than 20 years. Exclusion criteria were age at surgery > 18 years, other etiology (neurologic, syndromic, congenital).

After the first database review, 302 patients met the inclusion criteria and were selected to supply contact information (email or telephone number) using our hospital patient management software. Among them, 133 patients were contacted, and a questionnaire drafted in the form of a “Google Form” was mailed to patients along with an invitation to participate in the study, after a telephone conversation during which patient consent was collected. A total of 63 patients completed the form and were included in the study (Figure 1).

Through the questionnaire, we collected the following information: VAS back, VAS leg, Scoliosis Research Society 22 (SRS-22), Short Form 12 (SF-12), revision surgery rate, daily life aspects (pregnancy, work and sport activities).

Then, patients were further subdivided into two groups depending on the surgical procedure performed (Figure 2):

(1) Non-instrumented fusion according to Hibbs–Risser technique [5] (Group A): This technique is based on a meticulous fusion executed on each hemi-space, either on the concave side or the convex side of the curve.

The intervention, carried out following a median posterior surgical access, begins by identifying the supraspinous ligament, which is dissected longitudinally at the apex of the spinous apophysis and continuing to detach the periosteum from the two sides of the spinous process and, therefore, from the laminae, until reaching the transverse apophyses. Once the vertebral arches have been completely exposed and the capsular and ligamentous structures have been carefully eliminated, cortical bone is attacked by the chisel first removing the facet joints (inferior facet joint of the upper vertebra). In this way, a large quantity of autologous bone is obtained from the posterior structures (laminae, spinosa, transverse apophysis), which is prepared to obtain bone grafts that are reversed in sequence and applied at each level. In all cases, an iliac bone graft was applied to increase the fusion power.

The surgical treatment of scoliosis with the Risser technique involves preoperative correction (for a period of 3 months) and postoperative application of a cast (for a period of approximately 6 months).

(2) Instrumented posterior fusion according to Cotrel–Dubousset technique [6] (Group B): introduced in 1980s, this system uses double rods and multiple spinal posterior element fixation anchors. In our series, a hybrid construct involving lumbar pedicle screws and thoracic hooks was used. Pedicle screws were inserted using the freehand technique. All the instrumentations included a distal anchor by using four pedicle screws in the lower two vertebrae. Pedicle screws were applied in the lumbar spine and distal thoracic vertebrae (T9/T10). Instead, pedicle hooks were positioned in the proximal thoracic vertebrae with a cephalad direction. The hook was applied with the combination of a hook holder, a mallet, and a hook-pusher. In the convex side of the scoliotic curve, at the upper instrumented vertebra, a transverse process hook with a caudal direction was positioned to reach a stable anchor point. Screws at each level were applied alternatively on the concave and convex side of the scoliotic curve, but a greater density was usually performed on the concave side. The apical vertebra was always included in the instrumented vertebrae. The spinous process, supraspinous and interspinous ligament, and the other spine restraints were removed to facilitate the correction maneuvers. The laminae were fully and scrupulously decorticated. Bone graft obtained from decortication and bone removal was used for fusion, applying it directly to the posterior bone surfaces.

The scoliosis correction process began with the application of the first rod in the concave side of the main curve. The rods were previously accurately modeled to reproduce the correct sagittal shape of the instrumented spinal segment, paying attention to obtain the ideal thoracic kyphosis and lumbar lordosis. A balanced spine in the sagittal and coronal plane was a crucial goal to achieve; often, to prevent the remodeling of the prebent rods during correction, a hyper-kyphosis and hyper-lordosis were given when rods were modeled.

After bringing the rod closer to the screws, an initial correction was obtained by a segmental translation of the vertebrae toward the rod.

In practice, the rod was reduced into the reduction tabs to reach the screw head by using the setscrews.

Once the rods were engaged in all anchors, the surgeon and his assistant performed a global derotation of about 90° through the use of rod rotation instruments, in the direction of the concave side of the scoliotic curve, reaching the greatest degree of correction.

To obtain further correction and improve the deformity also on the axial plane, an additional segmental derotation was also performed.

When the patient was affected by a very stiff curve, additional correction maneuvers with segmental compression and distraction were applied [6].

For this study, we evaluated the functional outcomes of the entire cohort of patients and then divided them into the above groups and compared the results.

### Statistical Analysis

The statistical analysis was performed using R Software v4.1.1 (R Core Team, Vienna, Austria). Continuous variables distribution was assessed by Shapiro–Wilk test. According to the result of this test, comparisons between groups were performed using Student t-test or Wilcoxon rank-sum test, in case of normal and non-normal distribution, respectively. Differences in the proportion of categorical variables were assessed by Fisher’s exact test. *p*-Values < 0.05 were considered statistically significant.

## 3. Results

At the end of the inclusion and exclusion process, 63 patients respected inclusion criteria, sent the completed questionnaires, and were enrolled in the study for statistical analysis. Group A and B were, respectively, 42 and 21 patients. The mean age at surgery was 15.7 ± 1.8 years and the mean follow-up was 32 ± 7.3 years. When patients were interviewed via our questionnaire, the mean age was 47.5 ± 6.3 years. Mean age and follow-up among the two groups were different because the non-instrumented technique was older. The features of the patients are summarized in Table 1. The scoliotic curves were reevaluated from the radiographic images and were classified according to Lenke’s classification (Lenke 1–21 patients; Lenke 3–18 patients; Lenke 5–18 patients; Lenke 6–4 patients; Lenke 3–2 patients). On average, 10.3 levels were fused, from a minimum of 8 to a maximum of 14 levels.

Overall outcome measures (PROMs) showed good results in both groups, although 12 patients (19%) needed revision surgery, significantly more in the instrumented group (11.9% vs. 33%, *p* < 0.05).

The mean value of VAS back, VAS leg, Short Form 12 PCS, Short Form 12 MCS, and SRS-22 without group distinction resulted to be, respectively, 3.5 ± 3.11, 2.51 ± 2.7, 41.1 ± 11.8, 46.7 ± 9.8, and 3.5 ± 0.7. The average satisfaction score was 3.7 ± 1.2 out of a maximum value of 5. The groups comparison showed no significant differences in VAS back (*p* = 0.533), VAS leg (*p* = 0.520), SF-12 PCS (*p* = 0.901), SF-12 MCS (*p* = 0.694) as well as the SRS-22 (*p* = 0.804) (Figure 3 and Figure 4).

The general satisfaction score was 3.7 ± 1.2 out of 5. The group comparison showed no statistically significant differences in VAS back (*p* = 0.533), VAS leg (*p* = 0.520), SF-12 PCS (*p* = 0.901), SF-12 MCS (*p* = 0.694) as well as the SRS-22 (*p* = 0.804) (Table 2).

Regarding survival rate, the two groups were significantly different (*p*-value < 0.001), with the greatest difference within the first 5 years. In fact, the rate of non-reoperation was 97.6% (CI95%: 84.3–100.0%) in the non-instrumented group and 71.4% (CI95%: 47.1–86.0%) in the instrumented group at 5-year follow-up (Figure 5).

This was mainly due to implant issues.

Nevertheless, as stated before, a higher revision rate did not lead to worse long-term clinical results, as demonstrated by the comparable results between group A and group B in terms of pain, physical and mental state, and quality of life at long follow-up.

Overall, 87.3% of patients had stable jobs. The percentage was slightly lower in Group B than in Group A (85.7% vs. 88.1%). A successful pregnancy was achieved in 56% of all the patients: 59% in Group A and 50% in Group B (Table 3). A higher prevalence of cesarean sections compared to vaginal deliveries was assessed (21–65.6%–versus 11–34.4, respectively).

A total of 34 patients (54% of the entire cohort) used to practice sport activities before surgery (50% amateur, 44.1% competitive, and 5.9% professional). A total of 79.4% of them returned to sport in the postoperative and 61.7% at last follow-up.

However, 20.6% (7 patients) stopped their sport because of thoracic and/or low back pain, functional limitation, or different reasons.

Dividing patients according to the level and intensity of sport activity, based on the American Academy of Pediatrics Classification [7], 47% of patients resumed a medium or high-intensity sport (level 3 or higher) in the postoperative, and 27% at last follow-up (Table 4).

## 4. Discussion

The present study shows a series of young patients operated for adolescent idiopathic scoliosis with long-term clinical follow-up. Our evaluation of the data does not intend to compare the results of two surgical techniques used for the treatment of AIS (non-instrumented and instrumented fusion). In fact, it would be useless to compare two techniques that are so different and developed several years apart. The main purpose of the study is to show the long-term clinical results of surgery especially on the quality of life of these young patients with high-functional demands. To our knowledge, this is one of the largest patients’ series with such a long follow-up study of individuals surgically treated for AIS.

First, we confirmed the success and overall satisfaction of surgical treatment of scoliosis in patients with significant preoperative clinical alterations. The main indications for surgical treatment were AIS exceeding a certain degree of Cobb’s angle (45°–50°), failure of conservative treatment, or symptomatic AIS [8], with still a wide range of differences according to the surgeon’s preferences.

It should be noted that there is today no randomized or non-randomized trial-based evidence from prospective series with a control group comparing the outcomes of surgical to conservative treatments for patients affected by AIS and severe curves of over 45 degrees [9].

Akazawa et al. [10] compared 66 operated patients with 76 healthy age and sex-matched people with neither a history of spinal surgery nor spine deformity and found no statistical differences in back or leg pain, physical and mental health (SRS-22), and low back pain severity (RDQ) between patients and controls, indicating good long-term outcome of surgical treatment for AIS. Still, in Akazawa et al. function and self-image scores on the SRS-22 questionnaire were significantly lower in the AIS group than in the control group (function: 4.3 ± 0.6 and 4.7 ± 0.5 [*p* < 0.0001] and self-image: 3.0 ± 0.8 and 3.7 ± 0.5 [*p* < 0.0001], respectively). Another recent study by Farshad et al. [11] compared 16 operated patients with 16 matched patients with a conservatively treated AIS with a long-term follow-up (47 and 39 years, respectively, for the surgical and conservative group). They found no differences in functional scores (ODI score) but found a relevant smaller curve magnitude with surgical treatment (38° for surgery group vs 61° for conservative group at final follow-up, starting, respectively, from 48° and 40°). Ghandhari et al. [12], in a study on 42 patients and 5.6 years follow-up, found benefits about aesthetics, quality of life, disability, back pain, psychological well-being, and breathing function, but also alert about potential longer-term risks such as greater strain on unfused vertebrae, curvature progression, decompensation of the deformity, and degenerative disk disease.

The global functional results are still so good that, in the scientific literature, it is also confirmed that, after a spinal fusion for AIS, a full return to sport is generally allowed. Barile et al. [13], in their review of 2021, showed that a return to sport after surgery ranges from 6 to 18 months postoperatively, while operated patients can safely return to any sports. However, in some patients, especially after extremely long spine fusion, the loss of mobility could make it difficult for patients to play at the same level as preoperatively. According to Pepke et al. [14], 29.2% of a series of 33 patients operated of spinal fusion for AIS could return to the same level of preoperative sport activity. Many patients in this study who resumed sports postoperatively shifted from contact sports toward lower level and intensity sports activities. The extent of spinal fusion had no influence on the time to return to training and full sports-specific activity.

In our series, even many years after surgery, the maintenance of good clinical and functional scores for most of the patients is indicative of a high long-term satisfaction rate, improvement of clinical issues, and good overall quality of life. Despite the evidence of greater rate of surgical revision in instrumented group, good overall functional outcomes were found at last follow-up: SRS-22 = 3.5 ± 0.7, VAS back = 3.5 ± 3.11, VAS leg = 2.5 ± 2.7, SF-12 PCS = 41.1 ± 11.8, SF-12 MCS = 46.7 ± 9.8, indicating general good health, without statistical differences between two groups (*p* > 0.05) (Table 2).

With some surprise, our study demonstrates that even with the surgical technique that does not involve the use of instrumentation, the long-term functional outcomes are good and comparable to the most recent instrumented fusion technique. On the other hand, the higher rate of surgical revisions related to the use of implants in the instrumented fusion technique also does not appear to have a negative impact on long-term clinical follow-up in our series of patients.

In our series, the rate of non-reoperation was 97.6% (CI95%: 84.3–100.0%) in the non-instrumented group and 71.4% (CI95%: 47.1–86.0%) in the instrumented group at 5-year follow-up. The introduction of instrumentation increased the incidence of adverse event and need for revision, especially in the first years of use. In recent years, there has been also an increase in the need for revision in patients treated with the non-fusion technique. The need for revision is generally postponed till long-term follow-up in these patients of adult age, and it is linked to the progressive decompensation of the spine and degeneration of adjacent segment. However, as our results show, it does not always have an influence on clinical outcome.

General satisfaction score was 3.7 ± 1.2 out of a maximum value of 5; 56% of the women in our series had at least one successful pregnancy, and at the last follow-up 87% of the patients declared they were regularly employed. These data should help the surgeon to reassure parents and young patients during the decisional process about choosing the surgical treatment, a moment that could be very stressful for both [15]. Another good result is that, 32 years after surgery, only 20.6% of the operated patients declared to have stopped sport activity because of pain or other reasons, while 79.4% of them returned to sport in the postoperative and 61.7% at last follow-up.

The results of the present study are even more reassuring if we think that the surgical techniques have evolved considerably over the last two decades [16]. All screw instrumentations for posterior spine fusion are now performed in almost all cases (98.4%); major complication rates decreased over time (from 18.7% to 5.1% at two years follow-up); greater improvements were observed in satisfaction, back pain, function, and quality of life.

Our study has some limitations. First, it is a retrospective analysis and survey. The design of the study has potential recall bias. Even though the total number of patients is considerable if compared to other long-term studies in literature, the sample size is still quite small and does not allow a real statistical comparison between the two types of surgical techniques. However, as mentioned above, the primary objective is not to compare the results of two very different surgical techniques used for the treatment of AIS but to show the long-term clinical results of surgery especially on the quality of life of these young patients with high-functional demands. An important limitation of our series is the absence of radiographic findings; because of the difficulty in collecting pre- and postoperative radiographic images in patients surgically treated many years ago (with the risk of excluding further patients from the study), we preferred not to include the radiographic results in our outcomes, focusing only on functional results, which are more important for patient satisfaction.

## 5. Conclusions

Patients surgically treated for adolescent idiopathic scoliosis show good outcomes at long-term follow-up. Pain, function, physical and mental status, and overall satisfaction are good, both in non-instrumented and instrumented fusion techniques. Most patients resume high-level sport activity and carry out regular work activity. Despite the higher rate of surgical revision in the instrumented technique compared to the non-instrumented one, the long-term functional results are not significantly affected.

## Figures and Tables

**Figure 1 children-11-00052-f001:**
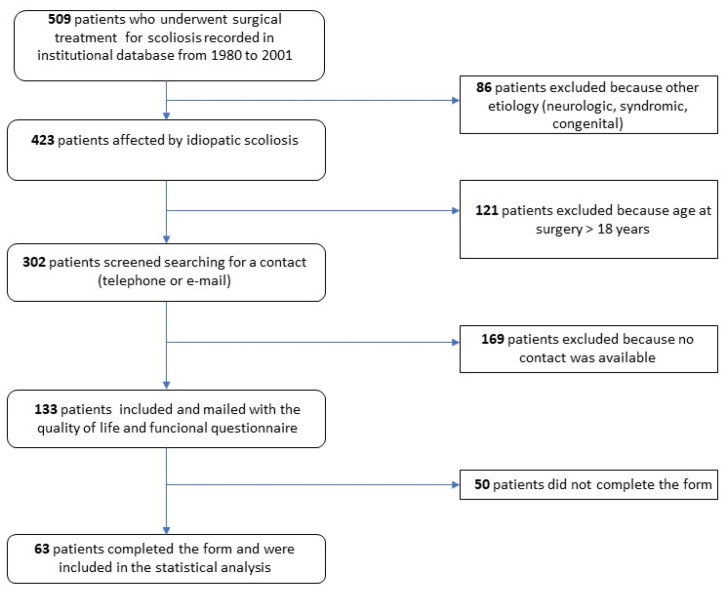
Flow diagram for inclusion and exclusion of patients in the study.

**Figure 2 children-11-00052-f002:**
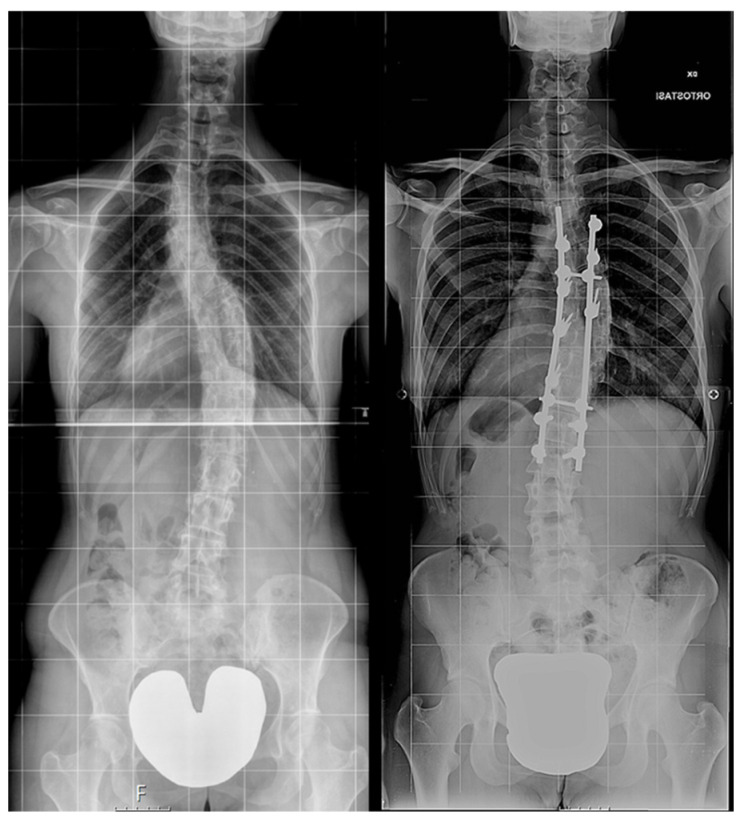
Examples of non-instrumented (**left**) and instrumented (**right**) long-term X-ray.

**Figure 3 children-11-00052-f003:**
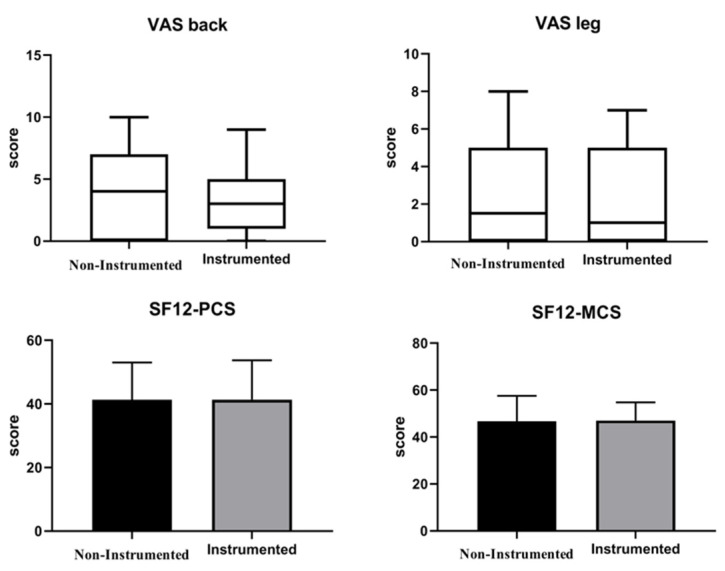
Functional outcomes reported by means of Tukey box and Whiskers plot with median (in-box line) and outliers (dots) in non-instrumented and instrumented fusions.

**Figure 4 children-11-00052-f004:**
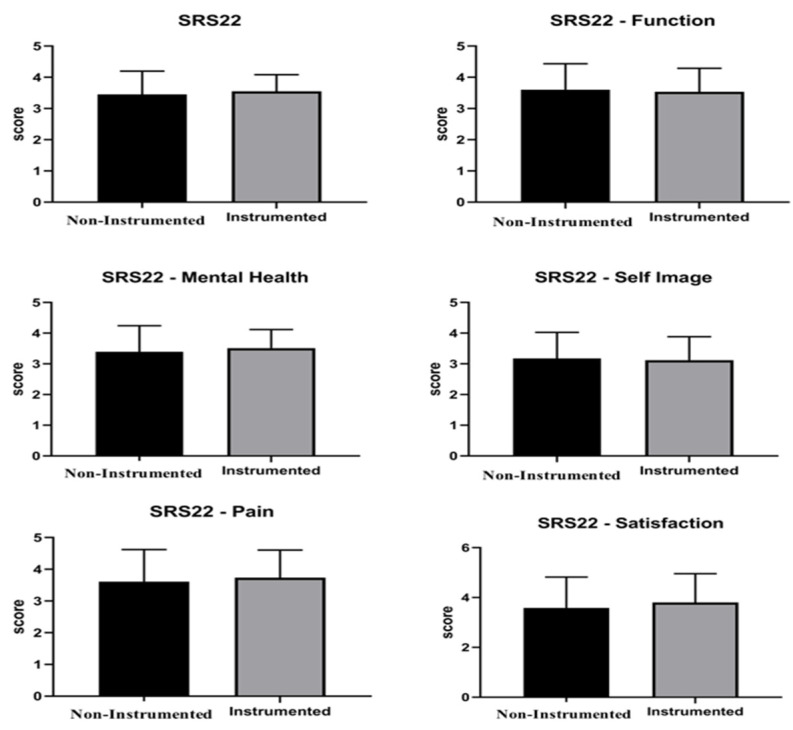
SRS22 general score and categories in non-instrumented and instrumented techniques.

**Figure 5 children-11-00052-f005:**
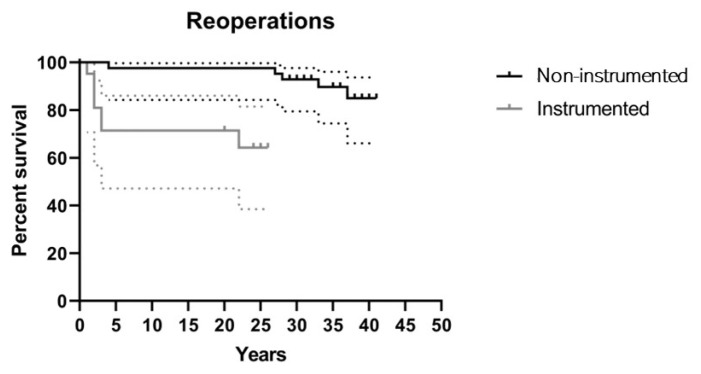
Survival curves depict the proportion of patients who did not undergo additional surgeries for each type of first intervention (non-instrumented and instrumented fusion).

**Table 1 children-11-00052-t001:** Data about patients grouped in non-instrumented (Group A) and instrumented (Group B) techniques; differences between all patients, non-instrumented, and instrumented techniques groups; mean [standard deviation].

	All Patients	Group A Non-Instrumented	Group B Instrumented	*p*
**Patients, *n***	63	42 (66.7%)	21 (33.3%)	-
**Mean Age, year**	47.5 ± 6.3	51.6 ± 5.1	37.3 ± 4.7	<0.001
**Age at surgery, year**	15.7 ± 1.8	15.6 ± 2	15.6 ± 2.1	0.538
**F/M**	57/6	39/13	18/3	0.391
**Follow-up, year**	32 ± 7.3	36.4 ± 3.9	23 ± 2.4	<0.001
**Revision rate**	12 (19%)	5 (11.9%)	7 (33%)	<0.05

**Table 2 children-11-00052-t002:** PROMs (patient-reported outcome measures) about patients grouped in non-instrumented (G1) and instrumented (G2) technique; differences between all patients, non-instrumented and instrumented technique groups; mean.

	All Patients	Non-Instrumented	Instumented	*p*
VAS back	3.5 ± 3.11	3.8 ± 3.3	3.1 ± 2.7	0.533
VAS leg	2.51 ± 2.7	2.7 ± 2.8	2.1 ± 2.7	0.520
SF-12 PCS	41.1 ± 11.8	41.3 ± 11.7	41.4 ± 12.3	0.901
SF-12 MCS	46.7 ± 9.8	46.7 ± 10.8	47 ± 7.8	0.694
SRS-22	3.5 ± 0.7	3.4 ± 0.7	3.5 ± 0.5	0.804
Function	3.6 ± 0.8	3.6 ± 0.8	3.5 ± 0.7	0.578
Pain	3.6 ± 0.9	3.6 ± 1	3.7 ± 0.9	0.703
Self Image	3.2 ± 0.8	3.2 ± 0.8	3.1 ± 0.8	0.994
Mental Health	3.4 ± 0.8	3.4 ± 0.8	3.5 ± 0.6	0.529
Satisfaction	3.7 ± 1.2	3.6 ± 1.2	3.8 ± 1.2	0.506

**Table 3 children-11-00052-t003:** Postoperative work activity and pregnancy.

Postoperative Work Activity and Pregnancy			
	ALL	NI-Technique	I-Techinque
Patients, n (%)	63	42 (66.7%)	21 (33.3%)
Work Activity–Stable job, n (%)	55 (87.3%)	37 (88.1%)	18 (85.7%)
Work Activity–Unemployed, n (%)	8 (12.7%)	8 (19%)	5 (23.8%)
All Female patients, n	57	39	18
Successful Pregnancy, n (%)	32 (56%)	23 (59%)	9 (50%)

**Table 4 children-11-00052-t004:** Level and intensity of sport activities, preoperatively, 1-year postoperative, and at the last follow-up.

	Preoperative	Postoperative	Last Follow-Up
All sports patients, n (%)	34	27 (79.4%)	21 (61.7%)
Level 1 (golf, bowling, walking)	/	/	5 (24%)
Level 2 (aerobic dancing, bicycling, jogging, swimming, tennis)	7 (21%)	14 (52%)	10 (48%)
Level 3 (fast running, weightlifting, high impact aerobic dancing, crew)	5 (17%)	5 (18.5%)	2 (9%)
Level 4 (gymnastic, volleyball, baseball, horseback riding, skating, skiing)	6 (18%)	3 (11%)	2 (9%)
Level 5 (basketball, boxing, football, soccer, martial arts, rugby)	16 (44%)	5 (18.5%)	2 (9%)
Amatorial	17 (50%)	18 (67%)	19 (91%)
Competitive	15 (44.1%)	8 (29%)	2 (9%)
Professional	2 (5.9%)	1 (4%)	0

## Data Availability

Data will be available on Zenodo.org.

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
