# Peer review of "Adolescent Idiopathic Scoliosis Surgery: Postoperative Functional Outcomes at 32 Years Mean Follow-Up"

_children, 2023, doi:10.3390/children11010052_

Round 1
Reviewer 1 Report
Comments and Suggestions for Authors
Dear authors,
Congratulations on this fine article, which sheds light on a very important topic in the treatment of spinal deformities.
I have the following comments:
Introduction:
Ll 53-54: there is no clear consensus of the cut-off for scoliosis surgery regarding cobb-angle. Therefore, you should not give an exact measure of 40° Cobb-angle as surgical indication. We usually recommend surgery for immature patients with a cobb angle of the structural curve which is progressive > 5° in the last year, when reaching >40° of cobb-angle and further growth is expected due to low Risser and/or Sanders stadium. There is also consensus of surgical treatment from 50° of cobb-angle in higher Sanders or Risser stadiums. Please give a little more information in this part of the introduction.
Methods:
L 108 – you mean chisel and not chiser?
L 141 – how do you obtain derotation forces when you use just hooks in the thoracic spine. This technique was introduces when more powerful pedicle screws where regularly use in the thoracic spine. Please clarify.
L142 – why do you only use compression and distraction in very stiff curves – this seems to be a standard reposition / correction tool
There is no information regarding ethics approval in the m
All in all, the article is very well written, comprehensible and answers the hypothesis put forward. Unfortunately, there were no more detailed questions regarding the birth process (spontaneous vs. caesarean section), which is interesting for many patients. Furthermore, radiological evaluations are unfortunately missing. The authors have mentioned and discussed this in the limitations.
The survival curve is also interesting. Please discuss this in more detail in the discussion. Although the non-fusion technique has fewer revisions in the first five years compared to the old fusion technique, it now appears to have a higher revision rate in the long term. However, this is not yet comparable due to the shorter follow-up in the fusion group. I would ask that this be dealt with accordingly in the discussion.
Author Response
Reviewer 1
R : Introduction: Ll 53-54: there is no clear consensus of the cut-off for scoliosis surgery regarding cobb-angle. Therefore, you should not give an exact measure of 40° Cobb-angle as surgical indication. We usually recommend surgery for immature patients with a cobb angle of the structural curve which is progressive > 5° in the last year, when reaching >40° of cobb-angle and further growth is expected due to low Risser and/or Sanders stadium. There is also consensus of surgical treatment from 50° of cobb-angle in higher Sanders or Risser stadiums. Please give a little more information in this part of the introduction.
A : Thank you for your suggestions. The authors revise the manuscript according to the suggestion.
R : Methods: L 108 – you mean chisel and not chiser?
A: Thank you for the suggestion the word was revised in the manuscript.
R : L 141 – how do you obtain derotation forces when you use just hooks in the thoracic spine. This technique was introduces when more powerful pedicle screws where regularly use in the thoracic spine. Please clarify.
A: Thank you for the comment. According to the original technique of Cotrel-Dubousset described in 1984 a general derotation is possible also with hybrid constructs, it was the innovation of the technique compared to the Harrington and Harrington-Luque, the presence of several anchor points gives the possibility to insert a pre-bent rod and perform a general derotation of the spine. Compared to the results obtained with the all-screw technique in a reduced amount and no segmental.
R: L142 – why do you only use compression and distraction in very stiff curves – this seems to be a standard reposition / correction tool
A: Thank you for the comment. In very elastic curve in young patients in our experience, the translation and general derotation maneuvers gives satisfactory correction, avoiding the neurologic risks linked to compression and distraction maneuvers. We generally prefer to use the compression and distraction when strictly necessary and when the amount of correction is not satisfactory.
R: There is no information regarding ethics approval in the manuscript
A: Thank you for the comment. The study was a retrospective study conducted only with telephonic and mail contact with the patients. Considering the noninvasiveness and retrospective nature of the study in our Institution and Ethical approval was not deemed necessary.
R: All in all, the article is very well written, comprehensible and answers the hypothesis put forward. Unfortunately, there were no more detailed questions regarding the birth process (spontaneous vs. caesarean section), which is interesting for many patients. Furthermore, radiological evaluations are unfortunately missing. The authors have mentioned and discussed this in the limitations.
A: Thank you for the comment.
R: The survival curve is also interesting. Please discuss this in more detail in the discussion. Although the non-fusion technique has fewer revisions in the first five years compared to the old fusion technique, it now appears to have a higher revision rate in the long term. However, this is not yet comparable due to the shorter follow-up in the fusion group. I would ask that this be dealt with accordingly in the discussion.
A: Thank you for the suggestion a paragraph has been add in the discussion section.
Reviewer 2 Report
Comments and Suggestions for Authors
Thanks for your interesting submission. The long-term outcome of AIS is a very important topic, particularly as we have become increasingly aggressive in it's treatment. I would note that patients become increasingly symptomatic after menopause (my own clinical observations), so a 40 year outcome study is also important.
A few notes/questions:
1. Keep the tables consistent - the headings should all be (for example) "Non-Instrumented" or all "NI"
2. The graphs add little value here - tables would be fine with mean / SD
3. Can you describe these curves more? Are they thoracic vs Thoracolumbar/lumbar? Were treatment levels noted?
One would imagine long-term natural history and return to sports are impacted by levels treated. This is a very important metric. I would imagine most of the curves are thoracic curves.
4. Can you elaborate on the pregnancy questions? Very interesting and patients would be very interested. Vaginal deliveries? Back pain during pregnancies?
5. You state that about 21% stopped sports due to pain. Can there be further elaboration here? Low back pain? Were these patients more likely to be fused lower?
We look forwards to more submissions on the long-term outcomes from this cohort.
Author Response
Reviewer 2
- R: Keep the tables consistent - the headings should all be (for example) "Non-Instrumented" or all "NI"
A: Thank you for your question. Table 3 has been modified making it consistent with the others
- R: The graphs add little value here - tables would be fine with mean / SD
A: Thank you for your note. In tables 1 and 2 the values are reported as mean and standard deviation. In tables 3 and 4, the use of percentage seems more effective.
- R: Can you describe these curves more? Are they thoracic vs Thoracolumbar/lumbar? Were treatment levels noted? One would imagine long-term natural history and return to sports are impacted by levels treated. This is a very important metric. I would imagine most of the curves are thoracic curves.
A: Thanks for this observation. We reevaluated the curves from the radiographic images and classified them according to Lenke's classification (Lenke 1, 21 patients; Lenke 3, 18 patients; Lenke 5, 18 patients; Lenke 6, 4 patients; Lenke 3, 2 patients). On average 10.3 levels were merged, from a minimum of 8 to a maximum of 14 levels. We have added this data to the "results" section. We did not perform statistical analysis on these subgroups because they were too small.
- R: Can you elaborate on the pregnancy questions? Very interesting and patients would be very interested. Vaginal deliveries? Back pain during pregnancies?
A: Thanks for this note. We assessed a higher prevalence of cesarean sections compared to vaginal deliveries (21 - 65.6% - versus 11 - 34.4, respectively). We have added this detail in the "results". We have not investigated back pain during pregnancy
- R: You state that about 21% stopped sports due to pain. Can there be further elaboration here? Low back pain? Were these patients more likely to be fused lower?
A: Thanks for this question. Patients who stopped sport activity due to pain reported thoracic and/or low back pain. The subgroup is small but in any case there appears to be no correlation with the extent of the fusion. We have detailed this in the results.